# Anti-inflammatory activity of diindolylmethane alleviates *Riemerella anatipestifer* infection in ducks

**Cherry P. Fernandez-Colorado**[1,2☉], **Paula Leona T. Cammayo**[2☉], **Rochelle A. Flores**[2], **Binh T. Nguyen**[2], **Woo H. Kim**[3], **Suk Kim**[2], **Hyun S. Lillehoj**[3], **Wongi Min**[2]*

1 Department of Veterinary Paraclinical Sciences, College of Veterinary Medicine, University of the Philippines Los Baños, College, Laguna, Philippines, 2 College of Veterinary Medicine & Institute of Animal Medicine, Gyeongsang National University, Jinju, Republic of Korea, 3 Animal Biosciences and Biotechnology Laboratory, Agricultural Research Service, United States Department of Agriculture, Beltsville, MD, United States of America

☉ These authors contributed equally to this work.
* wongimin@gnu.ac.kr

## Abstract

3,3'-Diindolylmethane (DIM) is found in cruciferous vegetables and is used to treat various inflammatory diseases because of its potential anti-inflammatory effects. To investigate effects of DIM in *Riemerella anatipestifer*-infected ducks which induce upregulation of inflammatory cytokines, ducks were treated orally with DIM at dose of 200 mg/kg/day and infected the following day with *R. anatipestifer*. Infected and DIM-treated ducks exhibited 14% increased survival rate and significantly decreased bacterial burden compared to infected untreated ducks. Next, the effect on the expression level of inflammatory cytokines (interleukin [IL]-17A, IL-17F, IL-6, IL-1β) of both *in vitro* and *in vivo* DIM-treated groups was monitored by quantitative reverse-transcription PCR (qRT-PCR). Generally, the expression levels of the cytokines were significantly reduced in DIM-treated splenic lymphocytes stimulated with killed *R. anatipestifer* compared to stimulated untreated splenic lymphocytes. Similarly, the expression levels of the cytokines were significantly reduced in the spleens and livers of DIM-treated *R. anatipestifer*–infected ducks compared to infected untreated ducks. This study demonstrated the ameliorative effects of DIM in ducks infected with *R. anatipestifer*. Thus, DIM can potentially be used to prevent and/or treat *R. anatipestifer* infection via inhibition of inflammatory cytokine expression.

## Introduction

Infection with *Riemerella anatipestifer*, referred to as riemerellosis, is often acute, contagious, and characterized by fibrinous exudates in the pericardial and hepatic cavities, meningitis, airsacculitis, caseous salpingitis, and septicemia [1]. The disease primarily affects domestic ducks, turkeys, geese, chickens, and other wild birds [1,2]. The mortality rate typically ranges from 5% to as high as 75%, depending on the virulence of the strain [1,3]. To date, at least 21 serotypes have been identified [3] and there is no significant serologic cross-protection between serotypes

**Data Availability Statement:** All relevant data are within the manuscript and its Supporting Information files

**Funding:** This research was supported by the Basic Science Research Program through the NRF of Korea, funded by the Ministry of Education (2018R1D1A1B07045179), and by Korea IPET through Agriculture, Food and Rural Affairs Convergence Technologies Program for Educating Creative Global Leader, funded by MAFRA-716002-7.

**Competing interests:** NO authors have competing interests

[4]. Although *R. anatipestifer* infection is contagious and thus poses a significant threat of economic losses in the duck industry worldwide [5,6], little progress has been made in elucidating the mechanism of host protective immunity against *R. anatipestifer* infection.

Considerable efforts to study the host immune response and molecular pathogenesis of *R. anatipestifer* have centered primarily on identifying virulence factors [7,8], immunogenic proteins [2,5,9], mutant strains that could serve as ideal live or attenuated vaccines [10–12], and common immunoreactive proteins between serotypes [13]. Furthermore, several studies have investigated the expression of cytokine and cytokine-related genes during *R. anatipestifer* infection. Expression levels of IL-6 and CCL19, cytokines related to inflammatory processes, are upregulated in the livers of ducks infected with *R. anatipestifer* [6]. Ducks vaccinated with inactivated *R. anatipestifer* plus levamisole as an adjuvant showed increased secretion of Th1-type (interferon [IFN]-γ and interleukin [IL]-2) and Th2-type (IL-4 and IL-10) cytokines and enhanced survival following challenge infection with a homologous *R. anatipestifer* strain [14]. Recent comparative analyses of the expression of immune-related genes between ducks and chickens revealed significantly higher IL-17A levels in both *R. anatipestifer*–infected ducks and killed *R. anatipestifer*–stimulated splenic lymphocytes [15,16]. Mice pre-treated with IL-17A or IL-23 prior to infection with *R. anatipestifer* at a sub-lethal dose exhibited increased bacterial burden and spleen weight compared to untreated infected mice [17].

The Th17 family of cytokines (IL-17A–F) has attracted attention due to its broad range of biological activities against pathogens [18,19]. IL-17A plays a particularly important role in host defense against infection with pathogens such as *Staphylococcus aureus* and *Citrobacter rodentium* [20], *Chlamydia muridarum* [21], and *R. anatipestifer* [15,16]. The critical involvement of Th1 and Th17 cells in the pathogenesis of various diseases has been demonstrated in a number of studies, and successful suppression of disease development is believed to involve neutralization or suppression of Th1 and Th17 cells [22–24]. Consequently, the use of certain anti-inflammatory agents, such as indoles (indole-3-carbinol [I3C] and 3,3'-diindolylmethane [DIM]), could be beneficial for the treatment of autoimmune diseases [25,26] and *Eimeria tenella* infection in chickens [27].

DIM is a natural bioactive compound derived from cruciferous (*Brassica*) vegetables, such as cabbage, cauliflower, Brussels sprouts, kale, turnips, and broccoli [28]. DIM has been shown to regulate immune responses and exhibit a broad range of biological activities in several disease models, including various cancers [29,30] and inflammatory diseases such as multiple sclerosis [26], colitis [25,31], and arthritis [32]. Moreover, DIM has shown potential effects against infections with enteric viruses [33] and *Staphylococcus aureus* [34,35]. The immuno-modulatory properties of DIM are associated with its ability to regulate multiple receptors [36,37] and signaling pathways, such as JNK, p38, NF-B, AP-1, and FAK [28,32,38,39], suggesting that the anti-inflammatory properties of DIM are associated with the regulation of T-cell differentiation and inhibition of Th1/Th17 cells. Therefore, this study was performed to investigate the potential use of DIM in alleviating the adverse effects of *R. anatipestifer* infection in ducks. We also examined the expression levels of inflammatory cytokines in DIM-treated splenic lymphocytes stimulated with killed *R. anatipestifer* and in the spleens and livers of DIM-treated *R. anatipestifer*–infected ducks in comparison with stimulated untreated splenic lymphocytes and infected untreated ducks.

## Materials and methods

### Animal ethics statement

All animal maintenance and experimental procedures were performed according to Gyeongsang National University Guidelines for the Care and Use of Experimental Animals

and approved by the Institutional Animal Care and Use Committee of Gyeongsang National University (GNU-170725-C0031). Humane endpoint criteria were set for all animals such that severe moribund animals exhibiting severe weight loss and tremors or unresponsive and unaware of stimuli were euthanized immediately by atlanto-occipital dislocation. All remaining animals were euthanized at specific time points post-inoculation as described below.

## Animals, infection, and treatment

One-day old Pekin ducklings were purchased from Joowon ASTA Ducks, Korea, and raised in wire cages in a temperature-controlled environment with constant light and unlimited access to antibiotic/anticoccidial-free feed and water. The birds were randomly assigned to four groups (n = 25/group) and housed in separate buildings: one group consisted of non-infected and untreated control birds; one group consisted of non-infected/DIM-treated birds; one group consisted of infected and untreated birds and one group consisted of infected/DIM-treated birds. The bacterium used in this study, *R. anatipestifer* serotype 7, was isolated from a commercial duck farm in Changwon, Gyeongnam Province, Korea, and serotyped at Chonbuk National University [40]. The isolate was grown on blood agar plates with 5% sheep blood (Asan Pharmaceutical, Korea), and a single colony was then cultivated in tryptic soy broth (BD Difco, USA) at 37˚C with vigorous shaking, as previously described [15,40]. Viable bacterial counts for the final challenge concentrations were determined by plating serial dilutions (10-fold) onto 5% sheep blood agar plates.

Two groups of ducks, at 2 weeks of age, were infected intramuscularly in the thigh muscle using a standard needle (26 gauge) with $5 \times 10^7$ colony forming units (CFUs) of *R. anatipestifer* serotype 7 in 200 μl of phosphate buffered saline (PBS). DIM (Sigma-Aldrich, St. Louis, MO, USA) (200 mg/kg) was administered orally daily beginning 1 day prior to infection and continuing throughout the experiment. Both the uninfected control and infected/untreated groups were administered an equivalent volume of PBS. Five birds in each group were euthanized via atlanto-occipital dislocation for tissue sample collection (i.e., liver and spleen) at 4 days post-infection (dpi). To determine animal susceptibility following *R. anatipestifer* infection, the survival rates of ducks (n = 25/group) were monitored for both control and treatment groups by recording the number of dead/moribund birds per day until day 10 post-infection.

## DIM preparation

DIM (≥98% purity-HPLC) was purchased from Sigma-Aldrich for cell treatment, and DIM capsules were obtained from BioPower (USA) for animal experiments. DIM was dissolved initially in dimethyl sulfoxide (DMSO) (Sigma-Aldrich) for *in vitro* studies as previously described [41], and mixed with feed for *in vivo* studies to obtain experimental concentrations (25 μM for cell treatment and 200 mg/kg for animal experiments).

## Isolation of duck splenic lymphocytes

The spleens were aseptically removed from 2-week-old healthy ducks and placed in normal Dulbecco's modified eagle's medium (HyClone, USA). Tissues were then minced and filtered using a sterile nylon mesh cell strainer (40 μm) (SPL, Korea). The cell suspension was diluted with an equal volume of PBS and carefully layered onto 10 ml of Ficoll-Paque PLUS solution (GE Healthcare, Sweden) for splenic lymphocyte isolation according to the manufacturer's instructions.

## MTT assay

The viability of isolated duck splenic lymphocytes was evaluated using an MTT assay, a colorimetric 3-(4,5-dimethylthiazol-2-yl)-2,5-diphenyltetrazolium bromide or thiazolyl blue staining method following the manufacturer's instructions. Cells were resuspended to $5 \times 10^6$ cells/ml and seeded in a 96-well plate, treated with different concentrations of DIM (0, 3.125, 6.25, 12.5, 25, 50, 75, 100 μM), and incubated for 24 h in a 41˚C incubator under 5% $CO_2$ After 24 h of incubation, MTT solution (1 mg/ml) (Sigma-Aldrich) was added, and the cells were further incubated for 4 h. DMSO (150 μl) was added to dissolve the formed dark blue formazan crystals in each well, and the absorbance at 540 nm was determined using a microplate reader. Cell viability was expressed as percent viability of treated cells versus the control (untreated, cultured cells), which was set at 100%.

## *In vitro* stimulation and DIM treatment of duck splenic lymphocytes

Splenic lymphocytes from 2-week-old healthy ducks isolated as described above were stimulated with heat-killed ($1 \times 10^6$ CFU/ml) *R. anatipestifer*, treated with DIM (25 μM), and incubated for 4, 8, or 24 h in a 41˚C incubator under 5% $CO_2$. Killed *R. anatipestifer* was prepared by boiling cells in a water bath at 100˚C for 5 min. Confirmation that *R. anatipestifer* cultures were killed prior to use in treating splenic lymphocytes was obtained by plating onto 5% sheep blood agar plates and monitoring for subsequent bacterial growth.

## Bacterial recovery

Tissue samples from livers (0.1 g) and spleens (0.05 g) were aseptically removed and homogenized separately in 500 μl of tryptic soy broth using tissue homogenizers. The homogenized samples were then serially diluted (10- or 100-fold) before plating onto 5% sheep blood agar plates. The plates were incubated at 37˚C under 5% $CO_2$ for 48 h. Viable bacterial colonies were counted to determine the number of colony forming units (CFUs)/ml.

## Quantitative reverse-transcription PCR (qRT-PCR)

Total RNA was extracted from tissue samples of five ducks from each group (control ducks, ducks infected with *R. anatipestifer* but untreated, *R. anatipestifer*–infected and DIM-treated ducks) as well as duck splenic lymphocytes stimulated with killed *R. anatipestifer*. RNA was extracted using RiboEx total RNA isolation solution (GeneAll, Korea). Prior to extraction, samples were homogenized using a grinder (Dalhan Sci., Korea) for tissues or a vortex for cells, according to the manufacturers' instructions. The extracted RNA was purified using an RNeasy Mini kit (Qiagen, Germany), treated with DNase I (Thermo Scientific, USA) to remove any contaminating genomic DNA, and quantified using an Optizen Nano Bio spectrophotometer (Mecasys, Korea). The treated RNA was used to synthesize a single-stranded cDNA using a QuantiTect Reverse Transcription Kit (Qiagen) and random hexamer primers. Quantitative real-time RT-PCR was performed in duplicate using a CFX96 real-time PCR system (Bio-Rad, USA) with the specific primers listed in Table 1. Gene expression was quantified using the comparative ΔΔCT method, with β-actin as a reference for normalization. The fold change in expression of each gene examined from *R. anatipestifer*–infected birds and *in vitro*–stimulated cells was calculated relative to the expression level in the same tissue or cells from uninfected or unstimulated samples, as previously described [15,42,49].

## Statistical analyses

The statistical significance of differences in data was determined by the Student's *t*-test or one-way ANOVA using InStat statistical software (GraphPad, USA). A *P*-value less than 0.05 was

**Table 1. Sequences of primers used for qRT-PCR analysis of cytokine expression.**

| Target Gene | Orientation | Sequence (5'-3') | Reference |
|---|---|---|---|
| IL-17A | Forward | ATGTCTCCAACCCTTCGT | [43] |
| | Reverse | CCGTATCACCTTCCCGTA | |
| IL-17F | Forward | CTGAGAGACTTAATGGAGACTG | [43] |
| | Reverse | AGAATCTGAACGGCTGATG | |
| IL-6 | Forward | TTCGACGAGGAGAAATGCTT | [44] |
| | Reverse | CCTTATCGTCGTTGCCAGAT | |
| IL-1β | Forward | TCATCTTCTACCGCCTGGAC | [44] |
| | Reverse | GTAGGTGGCGATGTTGACCT | |
| IFN-γ | Forward | CAACGCTCAACTACTCTC | [45] |
| | Reverse | TGTGGTTAATCTGTCCTTAG | |
| IL-10 | Forward | GAGATGATGCGGTTCTACAT | JN786941.1 |
| | Reverse | TTATGGTTTTGCTCCTCTTC | |
| β-Actin | Forward | GCTATGTCGCCCTGGATTTC | [46] |
| | Reverse | CACAGGACTCCATACCCAAGAA | |

considered to indicate a statistically significant difference. Data are expressed as mean ± standard error (SE).

## Results

### Effect of DIM on cell viability

MTT assays were carried out to investigate the effect of DIM on the viability of duck splenic lymphocytes (S1 Fig). Duck splenic lymphocytes were isolated from healthy ducks, cultured, and stimulated with different concentrations of DIM (0–100 μM). At a DIM concentration of 25 μM, cell viability was 93%, and this was the highest concentration of DIM that did not significantly affect the cells. Therefore, we used a DIM concentration of 25 μM for all subsequent *in vitro* experiments (Fig 1).

### Attenuation of *R. anatipestifer* infection by DIM treatment

Our previous studies indicated that bacterial burden during *R. anatipestifer* infection in ducks is the highest on day 4 post-infection [15,16]. Therefore, the bacterial load in the livers and spleens of infected/treated ducks was determined on day 4 post-infection. As shown in Fig 2A, bacterial burden was reduced significantly in the livers and spleens of infected/DIM-treated ducks compared to infected/untreated birds. Survival rate was monitored throughout the experiment, and *R. anatipestifer*–infected DIM-treated ducks exhibited a reduced mortality rate compared to untreated *R. anatipestifer*–infected ducks. Ducks infected with *R. anatipestifer* exhibited a 47% morality rate, whereas ducks infected with *R. anatipestifer* and treated with DIM exhibited a 33% morality rate, indicating a 14% increase in survival rate. DIM treatment alone had no effect on mortality (Fig 2B).

### Effect of DIM treatment on IFN-γ and IL-10 expression levels

Quantitative RT-PCR analysis was conducted to analyze the expression profiles of IFN-γ and IL-10 in splenic lymphocytes stimulated with heat-killed *R. anatipestifer* in the presence or absence of 25 μM DIM for 4, 8, and 24 h (Fig 3). Expression profiles of these cytokines were also examined in DIM-treated *R. anatipestifer*–infected ducks (Fig 4).

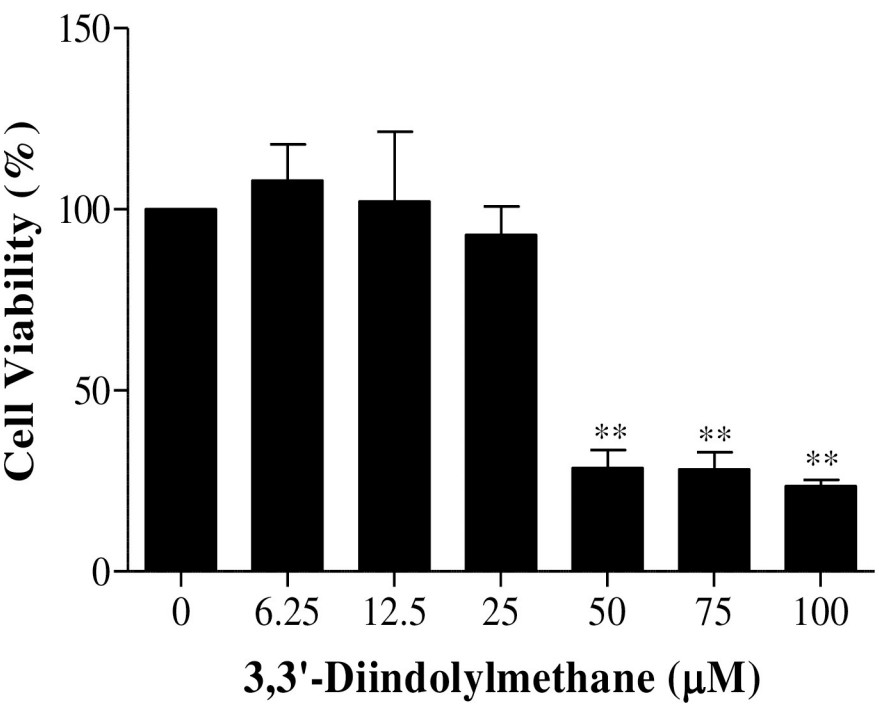

**Fig 1. Viability of splenic lymphocytes to 3,3'-Diindolylmethane.** Splenic lymphocytes were treated with different concentrations of DIM (0–100 μM) for 24 h, and cell viability was determined using an MTT assay. Cell viability is expressed as a percent of the viability of the control (untreated, cultured cells), which was set at 100%. Data represent the mean ± SE from three replicates of four independent experiments with similar results. Asterisks (**) indicate a significant difference relative to the control group ($P < 0.01$).

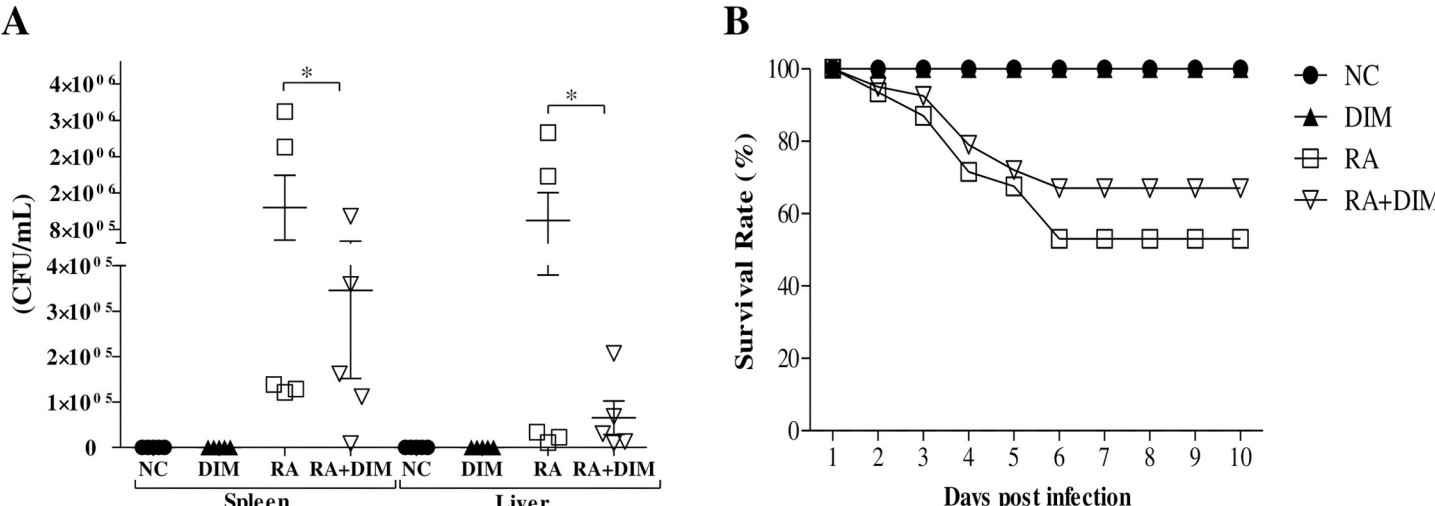

**Fig 2. DIM treatment attenuates *R. anatipestifer* infection in ducks.** (A) Bacterial load in the spleens and livers. Two-week-old ducks were inoculated intramuscularly with $5 \times 10^7$ CFU of *R. anatipestifer* serotype 7 and treated orally with DIM (200 mg/kg/day) from 1 day prior to infection throughout the experiment. Five ducks were sacrificed at 4 dpi, and the spleen and liver were aseptically removed for bacterial recovery. Data on bacterial recovery represent the mean ± SE of five birds and one representative of two independent experiments. $^*P < 0.05$ for comparison of the infected/untreated group (RA) with the infected/treated group (RA + DIM). (B) Survival rate of ducks (n = 25/group). The survival rate of control and treated ducks was recorded every day for 10 days. Data represent one representative of two independent experiments. NC, uninfected healthy control; RA, *R. anatipestifer*; DIM, 3,3'-diindolylmethane; CFU, colony formation unit.

Compared to unstimulated cultured (or control) splenic lymphocytes, IFN-γ transcript expression levels increased slightly, by 1.6- and 1.8-fold, in *R. anatipestifer*–stimulated splenic lymphocytes at 4 and 8 h, respectively, but not at 24 h. IFN-γ expression levels in stimulated and DIM-treated lymphocytes were significantly reduced at 4 and 8 h, whereas the level of IFN-γ expression was significantly enhanced in stimulated and treated lymphocytes at 24 h, compared to stimulated and untreated lymphocytes (Fig 3A). Levels of IL-10 transcripts were significantly increased both in *R. anatipestifer*–stimulated splenic lymphocytes and *R. anatipestifer*–stimulated/DIM-treated lymphocytes at all time points compared to control splenic lymphocytes(NC). Levels of IL-10 expression were significantly upregulated in stimulated and treated lymphocytes at 24 h, compared to stimulated and untreated lymphocytes (Fig 3B). DIM treatment alone had no effect on the levels of IFN-γ or IL-10 transcripts at any time point, except for the level of IFN-γ expression at 8 h, which was significantly enhanced (Fig 3A).

Compared to uninfected healthy control ducks, the levels of IFN-γ transcripts were significantly higher in the spleens of *R. anatipestifer*–infected birds as well as *R. anatipestifer*–infected/DIM-treated birds. IFN-γ expression levels in infected and treated birds were significantly upregulated in the spleen but not the liver, as compared with infected/untreated birds (Fig 4A). Expression levels of IL-10 were significantly increased in the spleens and livers of *R. anatipestifer*–infected birds compared to uninfected healthy controls. However, IL-10 expression levels in infected/treated birds were significantly reduced in the spleen and liver compared to infected/untreated birds (Fig 4B). DIM treatment alone had no effect on the expression of IFN-γ or IL-10 transcripts in either tissue (Fig 4).

## Downregulated expression of IL-17A and related cytokines following DIM treatment

DIM supressed Th17 cell differentiation, resulting in downregulation of IL-17A expression levels [25]. Our previous studies demonstrated upregulated expression of inflammatory cytokines, including IL-17A, in *R. anatipestifer*–infected ducks [15,42,47]. Thus, the expression

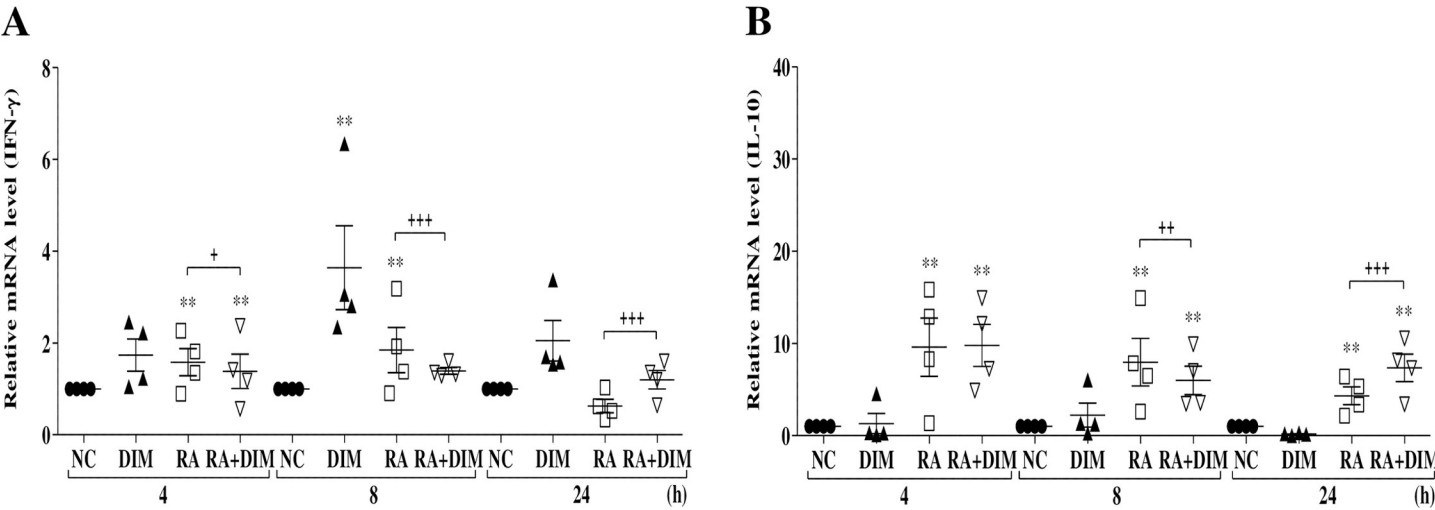

**Fig 3. Effect of DIM on IFN-γ and IL-10 expression in splenic lymphocytes.** Splenic lymphocytes collected from 2-week-old healthy ducks were stimulated with killed-*R. anatipestifer* and treated simultaneously with DIM (25 μM) for the indicated times. Samples were then subjected to qRT-PCR. The levels of IFN-γ (A) and IL-10 (B) mRNA were normalized to that of β-actin and calibrated using the expression levels of unstimulated cultured splenic lymphocytes (NC). Data are expressed as the mean ± SE from four independent experiments with duplicates. **$P < 0.01$ compared to NC. +$P < 0.05$, ++$P < 0.01$, and +++$P < 0.001$ for comparison of stimulated/untreated splenic lymphocytes with stimulated/treated splenic lymphocytes. RA, *R. anatipestifer*; DIM, 3,3'-diindolylmethane.

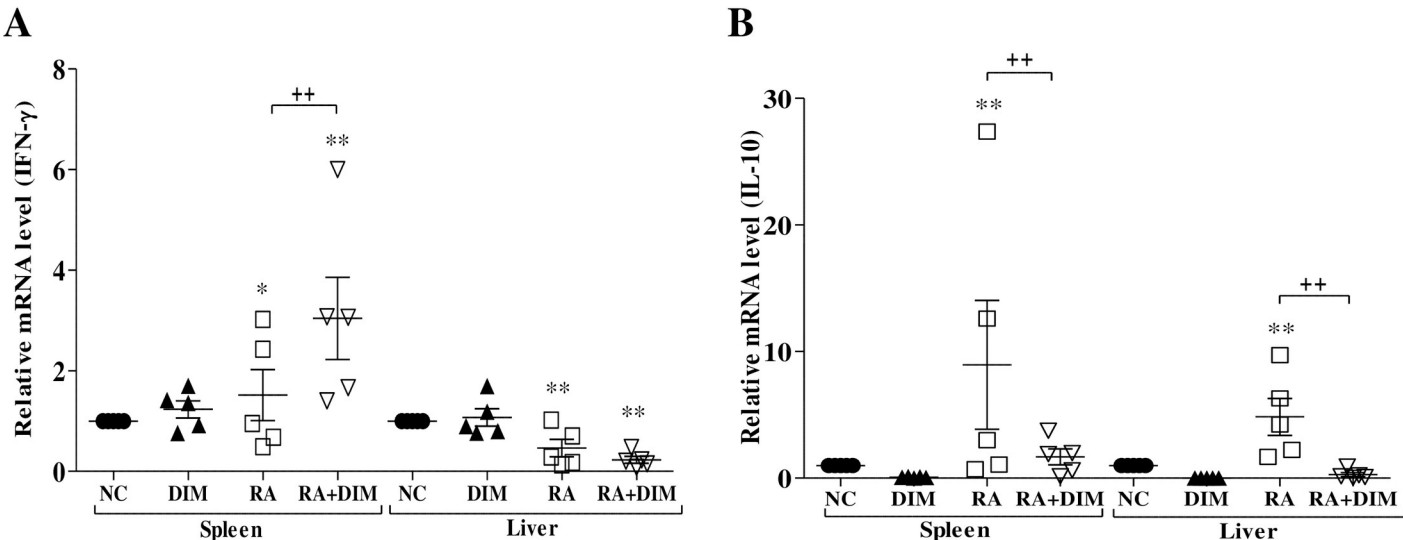

**Fig 4. Effect of DIM on IFN-γ and IL-10 expression in *R. anatipestifer*–infected ducks.** Two-week-old ducks were inoculated intramuscularly with $5 \times 10^7$ CFU of *R. anatipestifer* serotype 7 and treated orally with DIM (200 mg/kg/day) by gavage daily from 1 day prior to infection throughout the experimental period. Ducks (n = 5/group) were sacrificed on day 4 after infection. The spleen and liver were aseptically removed, homogenized using a grinder, and subjected to qRT-PCR. The levels of IFN-γ (A) and IL-10 (B) mRNA were normalized to that of β-actin and calibrated using the expression levels of uninfected/untreated healthy birds (NC). Data are expressed as the mean ± SE of five birds and one representative of two independent experiments. $^{*}P < 0.05$ and $^{**}P < 0.01$ compared to NC. $^{++}P < 0.01$ for comparison of the infected/untreated group with the infected/treated group. RA, *R. anatipestifer*; DIM, 3,3'-diindolylmethane.

profiles of inflammatory cytokines such as IL-17A, IL-17F, IL-6, and IL-1β were investigated by qRT-PCR in splenic lymphocytes stimulated with heat-killed *R. anatipestifer* in the presence and absence of 25 μM DIM for 4, 8, and 24 h (Fig 5). Expression profiles of these cytokines were also examined in DIM-treated *R. anatipestifer*–infected ducks (Fig 6).

As shown in Fig 5, expression levels of IL-17A and related cytokines were dramatically upregulated in both *R. anatipestifer*–stimulated lymphocytes and *R. anatipestifer*–stimulated/DIM-treated lymphocytes at all time points compared to cultured untreated control splenic lymphocytes. Interestingly, stimulated/DIM-treated lymphocytes exhibited significant down-regulation of IL-17A and IL-17F expression at 8 and 24 h compared with stimulated/untreated lymphocytes (Fig 5A and 5B). Levels of IL-6 expression were significantly reduced in stimulated/treated lymphocytes at 24 h compared to stimulated/untreated lymphocytes (Fig 5C). Similarly, IL-1β expression levels were significantly reduced in stimulated/treated lymphocytes at 8 and 24 h compared to stimulated/untreated lymphocytes (Fig 5D), although IL-6 and IL-1β expression levels were significantly increased only at 4 h (Fig 5C and 5D).

The *in vitro* results described above indicated that DIM has a negative regulatory effect on the expression of Th17-related cytokines. Hence, we further investigated whether the expression levels of inflammatory cytokines were downregulated in the spleens and livers of DIM-treated ducks. The expression of IL-17A and related cytokines was dramatically upregulated in *R. anatipestifer*–infected ducks compared with uninfected healthy control ducks. Compared to the *R. anatipestifer*–infected ducks, mRNA expression levels of Th17-related cytokines, including IL-1β, IL-6, and IL-17A were markedly reduced in the livers and spleens of all infected groups treated with DIM (Fig 6A, 6C and 6D). Although the level of IL-17F mRNA was markedly reduced in the liver, it was unchanged in the spleen (Fig 6B). Collectively, these results suggest that DIM treatment significantly suppresses the production of inflammatory cytokines both *in vitro* in stimulated duck splenic lymphocytes and *in vivo* in ducks infected with *R. anatipestifer*.

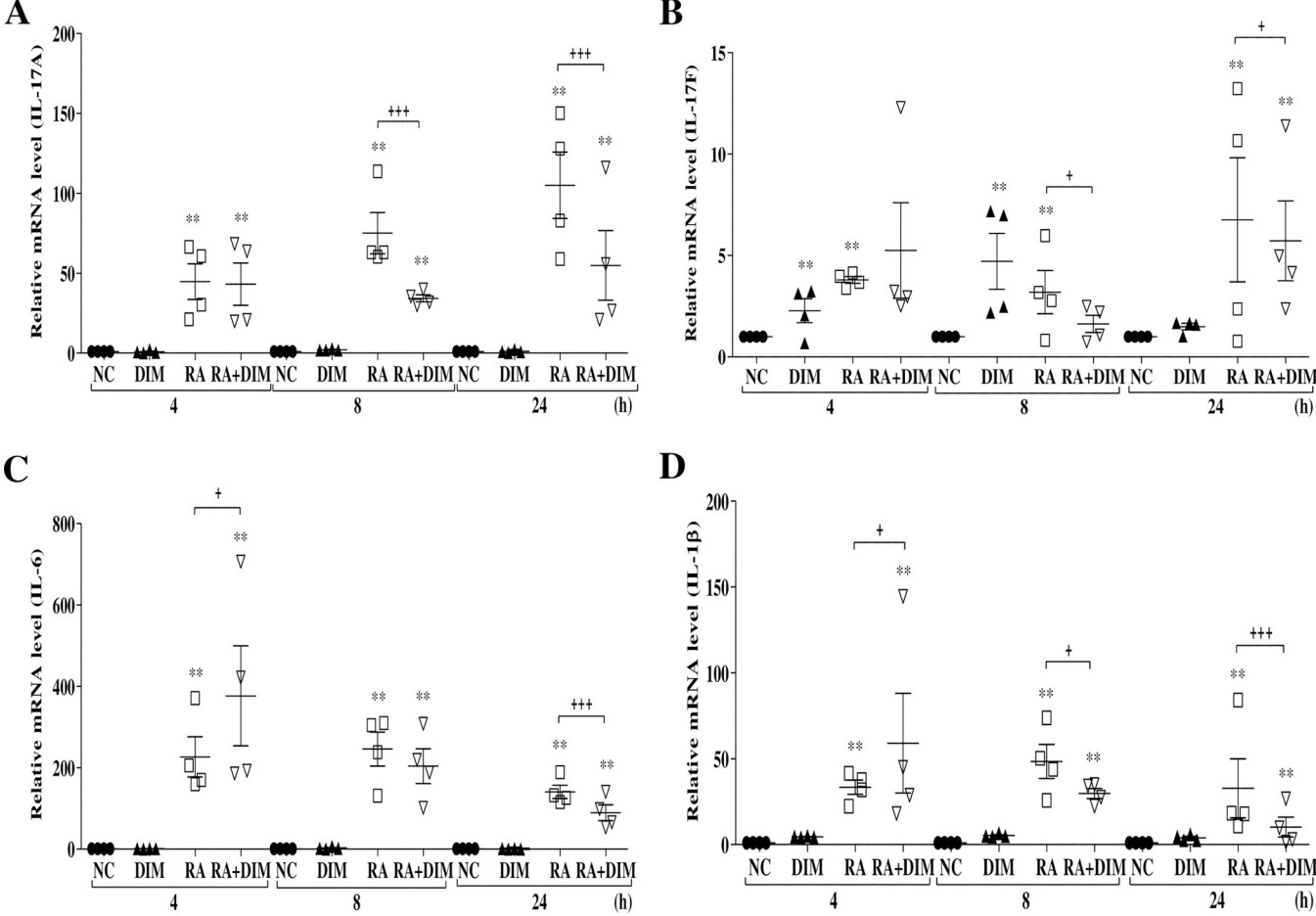

**Fig 5. Effect of DIM on the expression of IL-17A and related cytokines in splenic lymphocytes.** Splenic lymphocytes collected from 2-week-old healthy ducks were stimulated with killed-*R. anatipestifer* and treated simultaneously with DIM (25 µM) for the indicated times. Samples were then subjected to qRT-PCR. The levels of IL-17A (A), IL-17F (B), IL-6 (C), and IL-1β (D) mRNA were normalized to that of β-actin and calibrated using the expression levels of unstimulated cultured splenic lymphocytes (NC). Data are expressed as the mean ± SE from four independent experiments with duplicates. $^{**}P < 0.01$ compared to NC. $^{+}P < 0.05$ and $^{+++}P < 0.001$ for the comparison of stimulated/untreated splenic lymphocytes with stimulated/treated splenic lymphocytes. RA, *R. anatipestifer*; DIM, 3,3'-diindolylmethane.

## Discussion

Different species of birds exhibit differences in susceptibility to *R. anatipestifer* infection and differences in the elicited immune response, as demonstrated by our previous study, particularly in ducks and chickens [15]. Comparative analyses of the expression of immune-related cytokines revealed a significant association between upregulated expression of inflammatory cytokines such as IL-17A, IL-6, and IL-1β and *R. anatipestifer* infection in ducks but not chickens [15]. IL-17A is crucial for host protective immunity against various microbial pathogens, whereas Th17 cells expressing IL-17A are emerging as critical mediators of autoimmune diseases, thus increasing interest and research into the development of strategies to treat these autoimmune diseases. Recent studies examining the involvement of Th17 cells in the pathogenesis of various diseases revealed that neutralization or suppression of these cells suppresses disease development [22–24].

Few studies demonstrated that suppression of proinflammatory cytokine expression alleviates *R. anatipestifer* infection [42,44,48]. Chickens infected with *R. anatipestifer* exhibited

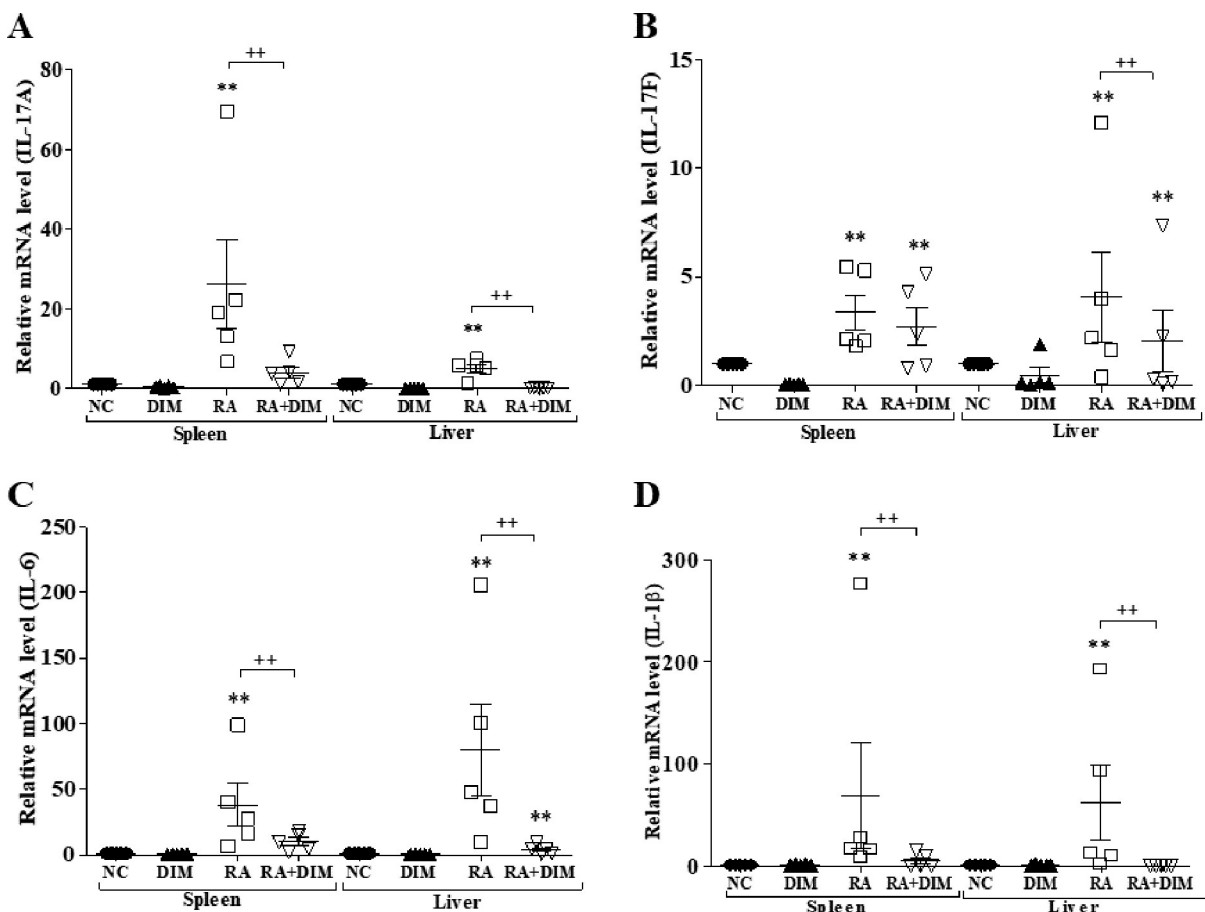

**Fig 6. Effect of DIM on the expression of IL-17A and related cytokines in *R. anatipestifer*–infected ducks.** Two-week-old ducks were inoculated intramuscularly with $5 \times 10^7$ CFU of *R. anatipestifer* serotype 7 and treated orally with DIM (200 mg/kg/day) by gavage daily from 1 day prior to infection throughout the experimental period. Ducks (n = 5/group) were sacrificed on day 4 after infection. The spleen and liver were aseptically removed, homogenized using a grinder, and subjected to qRT-PCR. The levels of IL-17A (A), IL-17F (B), IL-6 (C), and IL-1β (D) mRNA were normalized to that of β-actin and calibrated using the expression levels of uninfected/untreated healthy birds (NC). Data are expressed as the mean ± SE of five birds and one representative of two independent experiments. **$P < 0.01$ compared to NC. ++$P < 0.01$ for the comparison of the infected/untreated group with the infected/treated group. RA, *R. anatipestifer*; DIM, 3,3'-diindolylmethane; DPI, days post-infection.

lower susceptibility compared to infected ducks. This difference was attributed to upregulated expression of IL-4, the hallmark Th2 cytokine, in the livers and spleens of infected chickens, suggesting that IL-4 is involved in suppressing the expression of proinflammatory cytokines, including IL-17A [15]. Consequently, recombinant duck IL-4 significantly downregulated the expression of proinflammatory cytokines in *R. anatipestifer*–stimulated and IL-4–treated duck splenic lymphocytes [48]. Moreover, the use of an anti-inflammatory agent such as berberine not only increased the survival rate and decreased the bacterial burden, it also downregulated the expression of proinflammatory cytokines in ducks infected with *R. anatipestifer* compared with *R. anatipestifer*–infected ducks not treated with berberine [42]. These findings suggest that inhibiting the expression of proinflammatory cytokines, including IL-17A, can reduce the severity of *R. anatipestifer* infection in ducks. Here, we report ameliorative effects of DIM in *R. anatipestifer*–infected ducks.

Our present study demonstrated that DIM treatment significantly reduces the bacterial bur-den in the livers and spleens of *R. anatipestifer*–infected ducks (Fig 2A). It is interesting to note

that I3C, which is converted to DIM when given orally, exhibited antibacterial activity against clinical isolates of antibiotic-resistant organisms such as *Escherichia coli*, *S. aureus*, and *Pseudomonas aeruginosa* and antifungal activity against *Candida albicans* [49,50]. DIM derivatives have been shown to exert antibacterial effects against various gram-negative and gram-positive bacteria, indicating that drugs based on DIM could be efficacious against a number of infectious diseases [51]. The findings of our study suggest that DIM also exhibits antibacterial activity against *R. anatipestifer*. Generally, infection with *R. anatipestifer* is associated with a mortality rate ranging from 5–75%, as previously reported in ducks [15,42,52]. In the present study, DIM treatment of *R. anatipestifer*–infected ducks led to a 14% increase in the survival rate (Fig 2B). The survival rate was also significantly increased by DIM treatment in mice with hematopoietic injury induced by total body irradiation [53] and in rats pre-treated with DIM prior to radiation exposure [54].

Th1 and Th2 cytokines are reportedly negative regulators of Th17 immune responses [55]. Therefore, in this study, we also investigated whether DIM affects the expression of Th1 (IFN-γ) and Th2 (IL-10) cytokines. The expression of IFN-γ was significantly upregulated only at 24 h in *R. anatipestifer*–stimulated DIM-treated splenic lymphocytes and in the spleens of *R. anatipestifer*–infected DIM-treated ducks (Figs 3A and 4A). DIM has been shown to increase expression of the IFN-γ gene in MCF-7 human breast cancer cells, MDA-MB-231 cells, and Jurkat T cells. However, increased expression of the IFN-γ gene was not detected at all time points in a previous study [38]. Oral administration of DIM was shown to increase serum levels of IFN-γ in mice [33]. In contrast, DIM treatment did not result in a significant increase in IFN-γ expression during oxazolone-induced colitis in mice [25]. Peritoneal administration of DIM in mice did not affect serum levels of IFN-γ [33]. Expression of IFN-γ was significantly reduced in the colon of dextran sodium sulfate (DSS)-exposed mice treated with DIM [56]. IL-10 expression in present study was significantly upregulated at 24 h in *R. anatipestifer*–stimulated DIM-treated duck splenic lymphocytes in the present study (Fig 3B). In another study, the expression of IL-10 mRNA was significantly increased at 24 h in ConA-stimulated chicken splenic lymphocytes treated with DIM or I3C [27]. Furthermore, expression of IL-10 mRNA was significantly increased in the cecal tonsils of chickens treated daily with DIM or I3C for 14 days. IL-10 mRNA expression was frequently, but not always, upregulated in the cecal tonsils of *Eimeria tenella*–infected DIM-treated chickens compared to *E. tenella*–infected untreated chickens [27]. However, in an earlier study, Kim et al., [56] found that IL-10 expression was either unchanged or reduced in the colon of DSS-exposed DIM-treated mice, depending on the DIM concentration. In our study, IL-10 expression was reduced in the spleens and livers of *R. anatipestifer*–infected DIM-treated ducks compared with *R. anatipestifer*–infected untreated ducks (Fig 4B). Considered collectively, the discrepancies between the results of our study and others regarding IFN-γ and IL-10 expression may be associated with differences between *in vivo* and *in vitro* experiments, different infection target sites, and animal and disease models. In the present study, the physiologic relevance of IFN-γ and IL-10 expression in DIM-treated *R. anatipestifer*–stimulated splenic lymphocytes and *R. anatipestifer*–infected ducks is unclear; thus, further studies are necessary to better characterize the effects of DIM on the expression of these cytokines.

*In vitro* and *in vivo* analyses indicated significant reductions in the expression of inflammatory cytokines, including IL-17A, IL-7F, IL-1β, and IL-6, at 24 h after DIM treatment in duck splenic lymphocytes stimulated with *R. anatipestifer* and in the spleens and livers of *R. anatipestifer*–infected ducks. Similarly, recent studies have suggested that DIM exerts anti-inflammatory effects against autoimmune diseases via multiple signaling pathways, such as suppression of Th17 cell differentiation, which leads to a decrease in inflammatory cytokine expression [25,28,39,57]. The downregulated expression of IL-17A and IL-6 after DIM treatment was

similar to that observed in mice with autoimmune diseases such as colitis [56] and experimental autoimmune encephalomyelitis [58], in which IL-17A and IL-6 expression was also suppressed by DIM treatment. IL-6 expression was shown to be downregulated in inflamed ears of mice treated with topical DIM [59]. DIM significantly downregulated the expression of IL-6 and IL-1β in lipopolysaccharide-stimulated RAW264.7 murine macrophages [28]. In avian species, the expression of IL-17A mRNA was shown to be significantly downregulated at 24 h in mitogen-stimulated chicken splenic lymphocytes treated with DIM or I3C. Levels of IL-17A and IL-1F mRNA were significantly reduced in the cecal tonsils of chickens treated daily with DIM or I3C for 14 days [27]. In addition, in chickens challenged with the parasite *E. tenella*, DIM treatment resulted in a significant decrease Th17 cells, leading to downregulation of IL-17A expression in the later stages of [27]. These data suggest that DIM inhibits Th17-related cytokine production both *in vitro* and *in vivo* following *R. anatipestifer* stimulation or infection.

In conclusion, DIM treatment appears to suppress the development of riemerellosis by reducing the bacterial burden and the expression of inflammatory cytokines in tissues of *R. anatipestifer*–infected ducks, resulting in higher survival rates. Moreover, given the marked upregulation of inflammatory cytokine expression in both *R. anatipestifer*–stimulated splenic lymphocytes and *R. anatipestifer*–infected ducks in our previous studies [15,16,42,48], the results of the present study further suggest that inhibition of inflammatory cytokine expression could significantly reduce economic losses associated with *R. anatipestifer* infection in farmed ducks.

## Supporting information

**S1 Fig. Chemical structure of 3,3'-diindolylmethane (DIM).**
(DOCX)

## Author Contributions

**Conceptualization:** Woo H. Kim, Suk Kim, Hyun S. Lillehoj, Wongi Min.

**Data curation:** Cherry P. Fernandez-Colorado, Paula Leona T. Cammayo, Rochelle A. Flores.

**Formal analysis:** Cherry P. Fernandez-Colorado, Paula Leona T. Cammayo, Rochelle A. Flores.

**Investigation:** Cherry P. Fernandez-Colorado, Paula Leona T. Cammayo, Rochelle A. Flores.

**Methodology:** Binh T. Nguyen.

**Supervision:** Wongi Min.

**Writing – original draft:** Cherry P. Fernandez-Colorado.

**Writing – review & editing:** Wongi Min.

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
