## [Decision Letter · Decision Letter 0]

16 Oct 2020

PONE-D-20-21347

Anti-inflammatory activity of diindolylmethane alleviates Riemerella anatipestifer infection in ducks

PLOS ONE

Dear Dr. Min,

Thank you for submitting your manuscript to PLOS ONE. After careful consideration, we feel that it has merit but does not fully meet PLOS ONE’s publication criteria as it currently stands. Therefore, we invite you to submit a revised version of the manuscript that addresses the points raised during the review process.

Notably, you will see that both reviewers have suggested that you add additional information regarding your study.

We look forward to receiving your revised manuscript.

Kind regards,

François Blachier, PhD

Academic Editor

PLOS ONE

Journal Requirements:

Reviewers' comments:

Reviewer's Responses to Questions

**Comments to the Author**

1. Is the manuscript technically sound, and do the data support the conclusions?

Reviewer #1: Yes

Reviewer #2: Yes

2. Has the statistical analysis been performed appropriately and rigorously? 

Reviewer #1: Yes

Reviewer #2: Yes

3. Have the authors made all data underlying the findings in their manuscript fully available?

Reviewer #1: Yes

Reviewer #2: Yes

4. Is the manuscript presented in an intelligible fashion and written in standard English?

Reviewer #1: Yes

Reviewer #2: No

5. Review Comments to the Author

Reviewer #1: The article is good written, meets scientific principles, is systematic and clear.

Needs a little revision of the abstract and an explanation of the treatment groups. The results of the study will be more comprehensive, if the duck mortality data is added.

Reviewer #2: Line 32 - 36, the sentence is too lengthy and confusing, require to rephrase the sentence.

Does the author have any idea about the pharmacokinetic data of DIM in the animal and the potential metabolites that might have contributed to the anti-inflammatory effect?

6. PLOS authors have the option to publish the peer review history of their article (what does this mean?). If published, this will include your full peer review and any attached files.

Reviewer #1: No

Reviewer #2: No

---

## [Author Response · Author response to Decision Letter 0]

26 Oct 2020

PONE-D-20-21347: Anti-inflammatory activity of diindolylmethane alleviates Riemerella anatipestifer infection in ducks

We very much appreciate the reviewer’s incisive comments and thank Sir/Madam for a very thorough examination of our data and recommendations for improvement. Below are our point-for-point responses. We really hope these will meet your approval.

Comments from the editors and reviewers:

Reviewer #1

Comment: The article is good written, meets scientific principles, is systematic and clear. Needs a little revision of the abstract and an explanation of the treatment groups. The results of the study will be more comprehensive, if the duck mortality data is added.

Response: The Abstract was modified as “3,3’-Diindolylmethane (DIM) is found in cruciferous vegetables and is used to treat various inflammatory diseases because of its potential anti-inflammatory effects. To investigate effects of DIM in Riemerella anatipestifer-infected ducks which induce upregulation of inflammatory cytokines, ducks were treated orally with DIM at dose of 200 mg/kg/day and infected the following day with R. anatipestifer. Infected and DIM-treated ducks exhibited 14% increased survival rate and significantly decreased bacterial burden compared to infected untreated ducks. Next, the effect on the expression level of inflammatory cytokines (interleukin [IL]-17A, IL-17F, IL-6, IL-1β) of both in vitro and in vivo DIM-treated groups was monitored by quantitative reverse-transcription PCR (qRT-PCR). Generally, the expression levels of the cytokines were significantly reduced in DIM-treated splenic lymphocytes stimulated with killed R. anatipestifer compared to stimulated untreated splenic lymphocytes. Similarly, the expression levels of the cytokines were significantly reduced in the spleens and livers of DIM-treated R. anatipestifer–infected ducks compared to infected untreated ducks. This study demonstrated the ameliorative effects of DIM in ducks infected with R. anatipestifer. Thus, DIM can potentially be used to prevent and/or treat R. anatipestifer infection via inhibition of inflammatory cytokine expression”

In lines 216-219, the sentence was modified with the duck mortality data as “Ducks infected with R. anatipestifer exhibited a 47% morality rate, whereas ducks infected with R. anatipestifer and treated with DIM exhibited a 33% morality rate, indicating a 14% increase in survival rate. DIM treatment alone had no effect on mortality (Fig 2B)”

Comment: There are 3 group? 

Response: In lines 111-114, the sentence “The birds were randomly assigned to four groups (n = 25/group) and housed in separate buildings: one group consisted of infected and untreated birds; one group consisted of infected/DIM-treated birds; and one group consisted of non-infected control birds” was changed with “The birds were randomly assigned to four groups (n = 25/group) and housed in separate buildings: one group consisted of non-infected and untreated control birds; one group consisted of non-infected/DIM-treated birds; one group consisted of infected and untreated birds and one group consisted of infected/DIM-treated birds” 

Reviewer #2

Comment-1: Line 32 - 36, the sentence is too lengthy and confusing, require to rephrase the sentence.

Response: The indicated sentence “Generally, the expression levels of inflammatory cytokines (interleukin [IL]-17A, IL-17F, IL-6, IL-1β) were significantly reduced both in DIM-treated splenic lymphocytes stimulated with killed R. anatipestifer and in the spleens and livers of DIM-treated R. anatipestifer–infected ducks compared to stimulated untreated splenic lymphocytes and infected untreated ducks” was changed as “Next, the effect on the expression level of inflammatory cytokines (interleukin [IL]-17A, IL-17F, IL-6, IL-1β) of both in vitro and in vivo DIM-treated groups was monitored by quantitative reverse-transcription PCR (qRT-PCR). Generally, the expression levels of the cytokines were significantly reduced in DIM-treated splenic lymphocytes stimulated with killed R. anatipestifer compared to stimulated untreated splenic lymphocytes. Furthermore, the expression levels of the cytokines were significantly reduced in the spleens and livers of DIM-treated R. anatipestifer–infected ducks compared to infected untreated ducks”

Comment-2: Does the author have any idea about the pharmacokinetic data of DIM in the animal and the potential metabolites that might have contributed to the anti-inflammatory effect?

Response: There is still a lack or limited information on the effects and pharmacokinetic of DIM in poultry research. The pharmacokinetic data of DIM in ducks in particular have not been reported or studied elsewhere. However, several studies on the pharmacokinetic of DIM have been demonstrated in other animal models like mice or rats. 3,3’-Diindolylmethane (DIM) is a bioactive metabolite of indole-3-carbinol, a phytochemical produced by the breakdown of the glucosinolate glucobrassicin found in cruciferous vegetables. In mice, DIM is distributed to all tissues after oral administration, with the liver exhibiting the highest concentration. The highest concentration in the liver is reached about 25 minutes after oral administration of DIM and is gradually decreased over time(Anderton et al., Pharmacokinetics and tissue disposition of indole-3-carbinol and its acid condensation products after oral administration to mice. Clinical Cancer Research. 2004. 10. 5233-5241). The pharmacokinetic of DIM was also demonstrated in rats treated intravenously with 10mg/kg DIM and results showed that DIM has a fast metabolism and elimination of DIM with a terminal half-life of 0.737h (Wu et al., 2015 in Journal of Pharmacokinetics and Pharmacodynamics, DOI 10.1007/s10928-015-9421-5). 

Generally, DIM significantly decreased the release of nitric oxide (NO), prostaglandin (PG) E2, tumor necrosis factor alpha, interleukin (IL)-6, and IL-1beta which are related with inflammatory response (Cho et al., 3,3'-Diindolylmethane suppresses the inflammatory response to lipopolysaccharide in murine macrophages. The Journal of Nutrition. 2008. 138. 17-23). Furthermore, on the anti-inflammatory effects of IC3 and its metabolite, DIM, have been studied on CD4+ T cell population and in chickens infected with Eimeria tenella (Kim et al., Indole Treatment Alleviates Intestinal Tissue Damage Induced by Chicken Coccidiosis Through Activation of the Aryl Hydrocarbon Receptor. Frontiers in Immunology. 2019. 10. 560). In the mentioned study, it was suggested that DIM is a ligand for chicken aryl hydrocarbon receptor (AhR) where once activated by DIM and IC3, it will exhibit anti-inflammatory properties.

Q1. DIM has effects on the expression of these cytokines, also DIM was associated with signaling pathways such as JNK, p38, NF-κB, AP-1, and FAK. Which signal pathway is mainly? Also can you explain that how DIM kills bacteria on ducks?

Response: Several studies have demonstrated the association of signaling pathways in the expression of cytokines caused by indoles (DIM, IC3). IC3 has been reported to modulate pathogen-induced intestinal inflammation caused by the bacterium Citrobacter rodentium through aryl hydrocarbon receptor (AhR) activation (Kiss et al., 2011; Schiering et al., 2017). Recently, indoles (DIM, IC3) also exhibited the involvement of AhR in chickens infected with Eimeria tenella which resulted in an increase in the number of Treg cells and suppression of Th17 cells (Kim et al., 2019). Interestingly, the authors of the current study also performed mRNA expression analysis of MyD88, STAT3, TAK1 and NF-κB genes in R. anatipestifer-stimulated splenic lymphocytes as well as in DIM-treated stimulated splenic lymphocytes for 4, 8 and 24 h. Results showed that expression levels of MyD88, STAT3 and TAK1 were unchanged or statistically not significant (ns) between R. anatipestifer-stimulated and DIM-treated stimulated splenic lymphocytes after 24 h. However, expression level of NF-κB was downregulated after 24 h in DIM-treated stimulated splenic lymphocytes compared with R. anatipestifer-stimulated untreated splenic lymphocytes. Thus, we hypothesize that DIM is mainly associated with NF-κB and AP-1 signaling pathways as demonstrated in the following figure. However, we need to conduct more study to verify the exact mechanisms.

3,3’-Diindolylmethane (DIM) is a bioactive metabolite of indole-3-carbinol (I3C). Plasma and tissue I3C concentrations in mice administered by the oral route (250 mg/kg) are generally less than 24 ug/ml (Anderton et al., Pharmacokinetics and tissue disposition of indole-3-carbinol and its acid condensation products after oral administration to mice. Clinical Cancer Research. 2004. 10. 5233-5241). In addition, after oral administration to rats and human at dose of 200 mg/kg/DIM, the actual plasma concentrations of DIM are 150-230 ng/ml in rats and about 83 ng/ml in human (Paltsev, M. et al. Comparative preclinical pharmacokinetics study of 3,3′-diindolylmethane formulations: is personalized treatment and targeted chemoprevention in the horizon? The EPMA journal, 2013. 4. 25; Reed, G. et al. Single-Dose Pharmacokinetics and Tolerability of Absorption Enhanced 3, 3′-Diindolylmethane in Healthy Subjects. Cancer Epidemiology, Biomarkers and prevention, 2008. 17. 2619-2624). Based on these papers, we first tested if DIM directly kills R. anatipestifer at 0 μg/ml, 5 μg/ml and 10 μg/ml concentration on Tryptic Soy Agar (TSA) agar plates. It was 34-120 times higher than the actual plasma concentrations of DIM administrated with dose of 200 mg/kg.

Both concentrations showed similar numbers of bacteria compared to DIM-untreated control group. Accordingly, it is believed that DIM can kill bacteria by altering the immune status of ducks infected with R. anatipestifer. 

Q2. On the Fig.2A, in the RA infected group from liver, it displays lower bacterial load on three ducks, even close to zero. Did it mean that ducks clear bacterium by themselves, but has nothing to do with DIM treatment? Maybe you increase the number of ducks.

Response: LD50 was used in this study and the time point of significant changes, such as mortality and immune changes, are observed 4 days after R. anatipestifer infection. Thus, we expect 50% birds to be more vulnerable and the remaining 50% to be resistant. Fig. 2A indicated that the three ducks immunity either eliminated the bacteria themselves or the organ contained less than 100 bacteria. On the other hand, vulnerable birds in infected and DIM-treated group shown to have a reduced bacterial count. Although the figure legend mentioned that we performed two independent experiments in this study, at least four or five independent experiments were conducted and showed similar results. 

Q3. On the Fig.2B, only 14% survival rate is contributed by DIM, is there a signification on the clinical treatment? Maybe we increase infectious dose, whether the survival rate of ducks infected with RA and DIM treatment is lower same as survival rate of ducks infected with RA?

Response: Although there is no direct data for field trials, we believe that it can reduce the cost of damage (ex, mortality and antibiotics) caused by ducks infected with R. anatipestifer. When ducks were infected with high dose (10 times of LD50), ducks mostly died in both infected/DIM-treated group and infected/non-treated group. 

Q4. All cytokines detected are by RT-qPCR method, can you detect them by other methods, such as ELISA, WB?

Response: Compared to mice, there is a very limited or minimal studies on poultry research due to the availability of reagents and/or antibodies specifically for ducks. Hence, all cytokines in this study were detected by qRT-PCR. Given that the antibodies for cytokine detection are available commercially, we will try our best to do ELISA and WB.

Suggestion 1. Can you move Figure 1A to the supplementary results.

Response: Figure 1A was moved to the supplementary results as suggested by the reviewer.

Suggestion 2. The Fig.6D was wrong, can you correct it.

Response: Fig.6D was corrected as suggested by the reviewer in the revised manuscript.

---

## [Editor Report · Decision Letter 1]

29 Oct 2020

Anti-inflammatory activity of diindolylmethane alleviates Riemerella anatipestifer infection in ducks

PONE-D-20-21347R1

Dear Dr. Min,

We’re pleased to inform you that your manuscript has been judged scientifically suitable for publication and will be formally accepted for publication once it meets all outstanding technical requirements.

Kind regards,

François Blachier, PhD

Academic Editor

PLOS ONE

Additional Editor Comments (optional): The authors have given timely responses to the reviewer's comments, and have modified their manuscript accordingly.

---

## [Editor Report · Acceptance letter]

3 Nov 2020

PONE-D-20-21347R1 

Anti-inflammatory activity of diindolylmethane alleviates *Riemerella anatipestifer* infection in ducks 

Dear Dr. Min:

I'm pleased to inform you that your manuscript has been deemed suitable for publication in PLOS ONE. Congratulations! Your manuscript is now with our production department. 

Kind regards, 

on behalf of

Dr. François Blachier 

Academic Editor

PLOS ONE